# Visible Light-Enhanced Antibacterial and Osteogenic Functionality of Au and Pt Nanoparticles Deposited on TiO_2_ Nanotubes

**DOI:** 10.3390/ma13173721

**Published:** 2020-08-23

**Authors:** Kyoung-Suk Moon, Eun-Joo Choi, Ji-Myung Bae, Young-Bum Park, Seunghan Oh

**Affiliations:** 1Department of Dental Biomaterials and the Institute of Biomaterial and Implant, Wonkwang University School of Dentistry, Iksan 54538, Korea; ksemoon@hanmail.net (K.-S.M.); baejimy@wku.ac.kr (J.-M.B.); 2Department of Oral and Maxillofacial Surgery, Wonkwang University School of Dentistry, Iksan 54538, Korea; cejoms@wku.ac.kr; 3Department of Prosthodontics, Yonsei University School of Dentistry, Seoul 03722, Korea

**Keywords:** gold, platinum, titania nanotubes, visible light irradiation, antibacterial activity, osteogenic functionality

## Abstract

This study aimed at evaluating the visible light mediated antimicrobial and osteogenic applications of noble metal, such as gold (Au) and platinum (Pt) coated titania (TiO_2_) nanotubes (NTs). In this study, the Au and Pt nanoparticles (NPs) were deposited on anodized 100 nm TiO_2_ NTs by ion plasma sputtering. The Au and Pt NPs were mainly deposited on the top surface layer of TiO_2_ NTs and showed light absorbance peaks around the 470 and 600 nm visible light region used in this study, as seen from the surface characterization. From the results of antibacterial activity test, Au and Pt NPs that were deposited on TiO_2_ NTs showed excellent antibacterial activity under 470 nm visible light irradiation due to the plasmonic photocatalysis based on the localized surface plasmon resonance effect of the Au and Pt NPs. In addition, alkaline phosphate activity test and quantitative real-time PCR assay of osteogenic related genes resulted that these NPs promoted the osteogenic functionality of human mesenchymal stem cells (hMSCs) under 600 nm visible light irradiation, because of the synergic effect of the photothermal scattering of noble metal nanoparticles and visible light low-level laser therapy (LLLT). Therefore, the combination of noble metal coated TiO_2_ NTs and visible light irradiation would be expected to perform permanent antibacterial activity without the need of an antibacterial agent besides promoting osteogenic functionality.

## 1. Introduction

The success of an implant has been linked to functional aspects related to mastication and pronunciation; tissue physiological aspects, including excellent osseointegration without inflammation, and psychological aspects that satisfy aesthetics without causing pain or discomfort [1]. From this point of view, an implant would be a total failure if it fails to achieve functions and aesthetics for various reasons, such as material degradation, bacterial infection, improper osseointegration, and so on [2]. Bacterial infection is one of the common causes of peri-implantitis in dentistry, and it continues an inflammatory reaction between the implant and the surrounding bone that causes the failure of the implant [3].

Even though many techniques, including antibacterial agents, ozone treatment, and laser irradiation, have been used to prevent and treat bacterial infection, antibacterial treatment of implants already in the maxillofacial area is very difficult [4]. Many studies related to coating the implant surface with bioactive or biodegradable materials containing antibiotics have attempted to prevent bacterial growth [5,6,7]. However, it is difficult to overcome antibiotic resistance, side effects, and limitations of single use. Recently, alternative treatments without antibacterial agents, such as antimicrobial polymers [8], antibacterial Ag nanoparticles (NPs) [9,10], photodynamic therapy [11], and photocatalysis of titania (TiO_2_) [12,13,14] have been introduced due to increased multi-drug resistance of bacterial strains. Among these treatments, the photocatalytic antibacterial effect of TiO_2_ is expected to be very useful, since dental implants and medical implantable devices mainly use titania or titania alloy.

There have been many reports of additive doping to extend the limited photocatalytic effect of titania in the ultraviolet to visible light region [12,15,16,17]. Especially, the combination of noble metals (gold or platinum, Au or Pt, respectively), which have excellent biological compatibility, and TiO_2_ has been reported to show remarkable visible light photocatalytic activity [18,19,20,21,22]. In previous studies, we found four major absorbance peaks in the ultraviolet (UV), visible, and near infrared (NIR) regions and evaluated the visible–NIR laser-mediated release of antibacterial agent and the accompanying antibacterial activity of Au-coated TiO_2_ nanotubes (NTs) [23,24]. There are few studies that relate to antimicrobial activity using plasmonic photocatalysis and the photothermal scattering effect of noble metal-coated TiO_2_ NTs without antibacterial agents [25,26,27]. Additionally, the research relating to the combined effect of visible light and noble metal coated TiO_2_ NTs on the osteogenic functionality of human mesenchymal stem cells was not conducted in detail.

Therefore, in this study, we prepared Au-deposited TiO_2_ NTs (Au–TiO_2_ NTs) and Pt-deposited TiO_2_ NTs (Pt–TiO_2_ NTs) by ion plasma sputtering and estimated their surface properties. We also evaluated the visible light-promoted antibacterial activity and osteogenic capability of Au–TiO_2_ NTs and Pt–TiO_2_ NTs.

## 2. Materials and Methods

### 2.1. Materials Preparation of Au–TiO_2_ NTs and Pt–TiO_2_ Nanotubes

As reported previously [23], 100 nm diameter TiO_2_ NTs were prepared by anodizing a pure titanium sheet (250 μm thick, 5 × 5 cm^2^, 99.5%; Hyundai Titanium Co., Incheon, Korea) in a 0.5 *w*/*v*% hydrofluoric acid (48 *w*/*v*%; Merck, Kenilworth, NJ, USA) aqueous solution at 20 V for 30 min. The anodized TiO_2_ NTs were crystallized by heating at 400 °C for 3 h in air (heating and cooling rate = 1 °C/min). Au–TiO_2_ NTs and Pt–TiO_2_ NTs were prepared by ion plasma sputtering in an E-1030 unit (Hitachi Co., Tokyo, Japan) for various periods (1, 2, and 3 min) to determine the experimental condition with maximum light observance in the visible light region (400–700 nm). Uncoated TiO_2_ NTs were used as a control.

### 2.2. Surface Characterization

The morphological structure, elemental analysis, and optical properties of Au–TiO_2_ NTs, and Pt–TiO_2_ NTs were determined by field-emission scanning electron microscope (FE-SEM; S4800; Hitachi & Horiba Co., Tokyo, Japan), spherical aberration corrected scanning transmission electron microscope (STEM-Cs; JEM-ARM200F; JEOL, Tokyo, Japan), energy dispersive X-ray spectroscopy (EDX; Oxford Instruments Analytical 7582, Abingdon, UK), and diffuse reflectance UV–Vis–NIR spectrophotometry (SolidSpec-3700; Shimadzu Co., Kyoto, Japan). Based on the data from the FE-SEM images, the diameter, length, and aspect ratio of deposited NPs were calculated by image analysis software (ImageJ; Version 1.53b, NIH, Bethesda, MD, USA).

### 2.3. Antibacterial Activity Test

*Staphylococcus aureus* (*S. aureus*; ATCC 25923, Manassas, VA, USA) strain was used in order to evaluate the visible light-mediated antibacterial activity of Au–TiO_2_ NTs and Pt–TiO_2_ NTs. After incubation for 24 h, a cultured *S. aureus* suspension was diluted to a concentration of 1 × 10^5^ colony-forming unit (CFU)/mL. The experimental specimens (1 × 1 cm^2^) were inoculated with 500 µL of diluted bacteria solution, and the bacteria cultured in a 37 °C incubator. After 24 h of incubation, lab-fabricated visible light emitting diode (LED) lights (470 nm and 600 nm, power density = 5.5 mW/cm^2^, distance from the specimen = 1 cm) were used to irradiate the bacteria that were cultured on the specimen for 30 min. After visible light irradiation, 100 μL of the bacteria on the specimen was collected and diluted in phosphate buffered saline (PBS) solution. Subsequently, 100 μL of the diluted bacteria solution was inoculated on a 100 mm diameter agar plate and incubated for 24 h at 37 °C. After 24 h of incubation, the resulting CFU per unit volume was measured by visually inspecting the agar plate.

### 2.4. The Culture of Human Mesenchymal Stem Cells

Human mesenchymal stem cells (hMSCs) were obtained from Lonza Corporation (Basel, Switzerland). 4–6 passages of hMSCs were used in this study. The cell growth media of the hMSCs was Alpha-modified Eagle’s minimum essential medium (Invitrogen, Karlsbad, CA, USA) with 10% FBS (Invitrogen, CA, USA) and 1% antibiotics (Invitrogen, CA, USA). Osteogenic differentiation media of hMSCs were prepared by adding 10 mM β-glycerolphosphate (Sigma, St. Louis, MO, USA), 150 μg/mL ascorbic acid (Sigma, MO, USA), and 10 nM dexamethasone (Sigma, MO, USA) to cell growth media. All media were replaced every three days.

### 2.5. Calcein AM/EthD-1 Staining of hMSCs and MTT Cell Viability Assay

Live/dead staining using calcein acetoxymethyl (calcein AM, Invitrogen, CA, USA) and ethidium homodimer-1 (EthD-1, Invitrogen, CA, USA) dyes was done to determine the cell viability in hMSCs cultured on Au–TiO_2_ NTs and Pt–TiO_2_ NTs. The specimens (1 × 1 cm^2^) in each 12-well plate were seeded with hMSCs to a density of 30,000 cells/well. After 24 and 48 h of additional incubation, the specimen was washed twice while using PBS aqueous solution, and 500 µL of PBS solution with 2 µM calcein AM and 4 µM EthD-1 added. The stained cells were examined under an inverted fluorescence microscope (CKX41, Olympus Co., Tokyo, Japan).

An MTT assay was conducted to investigate the proliferation rate of hMSCs that were cultured on Au–TiO_2_ NTs and Pt–TiO_2_ NTs [28]. The autoclaved specimens (1 × 1 cm^2^) were placed on a 24-well plate, and hMSCs were seeded to each well to a density of 10,000 cells/well. After 24 and 48 h of incubation, the specimens were washed with PBS, and 1 mL MTT dye (Sigma, USA) was added to each well. After 4 h of incubation in a 5% CO_2_ incubator, 1 mL dimethyl sulfoxide (Sigma, USA) was added to each well, and the 24-well plate shaken for 30 min. The absorbance of each solution was measured at 540 nm using an ELISA microplate reader (SpectraMax 250, Thermo Electron Co., Waltham, MA, USA).

### 2.6. Morphological Evaluation of hMSCs with Visible Light Irradiation

The morphological change of hMSCs attached to Au–TiO_2_ NTs and Pt–TiO_2_ NTs before and after 470 and 600 nm visible light irradiation were observed by FE-SEM. The autoclaved specimens (1 × 1 cm^2^) were placed in a 24-well plate, and hMSCs were seeded to each well to a density of 3000 cells/well. After 24 h of incubation, the specimen was irradiated with 470 and 600 nm visible light (distance = 1 cm) for 30 min. After this, the specimen was washed twice using PBS, and 2.5% glutaraldehyde (Sigma, USA) was added to the specimen in order to fix the cells. After 1 h of reaction with glutaraldehyde at 4 °C, the fixed cells were sequentially dehydrated with ethanol to a 50%, 70%, 90%, and 100% concentration. Subsequently, the dehydrated cells were dried using a critical point dryer (CPD 030, Bal-Tec AG, Schalksmühle, Germany), and the filopodia of dried hMSCs observed by FE-SEM.

### 2.7. Alkaline Phosphatase (ALP) Activity Assay

The hMSCs were seeded onto the experimental specimen in a 24-well plate to a density of 20,000 cells/well. After 1 h of incubation, visible light irradiation (470 and 600 nm) was performed for 30 min. Additional visible light irradiation and fresh osteogenic media change was conducted every three days. After one and two weeks of cultivation, the cultured cells were collected and lysed in a lysis buffer solution (25 mM Tris, pH 7.6, 150 mM NaCl, 1% NP-40). After 30 min of lysis, 200 µL of para-nitrophenyl phosphate (pNPP; Sigma, MO, USA) was added to 50 µL cell lysate and reacted at 37 °C. After 30 min of p-NPP reaction, 50 µL of 3 N NaOH (Sigma, MO, USA) was added to the solution to stop the reaction. After measuring the absorbance at 405 nm using the microplate ELISA reader, the enzyme activity per unit protein content was calculated by dividing the absorbance value by the average value of the total protein amount. Four specimens were collected at a time to form one sample in order minimize the experimental errors. The number of samples used in the statistical analysis is three.

### 2.8. Quantitative Real-Time PCR Assay

The culture conditions and visible light irradiation procedure of hMSCs in the real-time PCR assay were the same as those for the alkaline phosphatase activity assay. After one and two weeks of cultivation, the total RNA of the cells on the experimental specimen were extracted using Trizol (Sigma, USA) reagent, and reverse-transcribed into cDNA using a Maxima First Strand cDNA Synthesis kit for RT-qPCR (Thermo Scientific, MA, USA). Real-time PCR analysis was done using Taqman^®^ Gene Expression Assays (Applied Biosystems, MA, USA). The specifications of the Taqman^®^ PCR primer were: GAPDH (Hs99999905_m1, Amplicon length: 122), ALP (Hs01029141_g1, Amplicon length: 71), OPN (Hs00960942_g1, Amplicon length: 63), and BSP (Hs00173720_m1; Amplicon length: 95). Real-time PCR measurement was carried out using a Taqman^®^ Universal PCR Master Mix solution and Applied Biosystems StepOne Real-Time PCR System (Applied Biosystems, Foster, CA, USA). 1 μL of the total volume of 20 μL cDNA samples was analyzed for the gene of interest while GAPDH was analyzed for the house-keeping gene. Four specimens were collected at a time to form one sample to minimize experimental errors. The number of samples used in the statistical analysis is three.

### 2.9. Data Analysis

All data were expressed as means ± standard deviations. The data for MTT assay, antibacterial activity, ALP activity, and quantitative real-time PCR assay were statistically analyzed by one-way analysis of variance (IBM SPSS Statistics 23.0; IBM, New York, NY, USA) and post-hoc Duncan’s multiple range tests. Differences were considered to be significant if the *p* values were less than 0.05.

## 3. Results

### 3.1. Suface Characteristics

#### 3.1.1. Field Emission Scanning Electron Microscope (FE-SEM)

Figure 1 shows FE-SEM images of Au–TiO_2_ NTs and Pt–TiO_2_ NTs after 1, 2, and 3 min of coating; and the diameter, height, and aspect ratio of Au and Pt NPs deposited on TiO_2_ NTs under various sputtering durations are listed in Table 1. In the 50 sputtered NPs observed by FE-SEM (Table 1), the diameter and height of Au NPs increased up to 18.29 ± 6.44 nm and 30.77 ± 3.28 nm after 3 min of sputtering. Additionally, the diameter and the height of Pt NPs increased up to 24.23 ± 4.40 nm and 48.24 ± 6.41 nm after 3 min of Pt sputtering durations. The aspect ratio of Au NPs and Pt NPs increased to 1.68:1 and 1.99:1, respectively, after 3 min of sputtering. After coating with Au for 2 min and with Pt for 3 min, agglomerated metal debris were detected on TiO_2_ NTs, as shown in Figure 1.

#### 3.1.2. Diffuse UV-Vis-NIR Spectroscopy

Figure 2 shows diffuse UV–Vis–NIR spectra of Au–TiO_2_ NTs and Pt–TiO_2_ NTs after various sputtering periods. The electron absorption spectra of all Au–TiO_2_ NTs and Pt–TiO_2_ NTs showed four main peaks at the following wavelength ranges: 350–400 nm, 400–500 nm, 550–650 nm, and above 800 nm. Two absorption peaks at wavelength ranges of 400 to 500 nm and 550 to 650 nm are equivalent to the peaks due to the plasmonic photocatalysis based on the localized surface plasmon resonance (LSPR) effect and the photothermal scattering of deposited Au and Pt NPs, respectively [29]. From the results of diffusive UV–Vis–NIR spectrophotometry, 1 min of Au sputtering and 2 min of Pt sputtering were selected in order to conduct additional characterization, antibacterial activity, and osteogenic functionality tests using a 470 and 600 nm visible LED light device.

#### 3.1.3. Transmission Electron Microscope (TEM)

Figure 3 shows vertical TEM images of Au–TiO_2_ NTs and Pt–TiO_2_ NTs (after 1 min of Au and 2 min of Pt sputtering). From the vertical TEM view of Au–TiO_2_ NTs and Pt–TiO_2_ NTs, most Au and Pt NPs were detected in the uppermost layer of TiO_2_ NTs—not in their inner surface. Additionally, 17.73 Au and 24.29 wt% Pt was detected in the mapped area of TEM images based on EDX analysis.

#### 3.1.4. X-ray Photoelectron Spectroscopy (XPS)

Figure 4 shows the XPS spectra of Au–TiO_2_ NTs and Pt–TiO_2_ NTs after 1 min of Au coating and 2 min of Pt coating. The main peaks of Ti 2p and O 1s shifted to lower binding energies due to the reduction of Ti species, as the Au and Pt NPs were coated on the surface of TiO_2_ NTs. In addition, the two main peaks detected from Au–TiO_2_ NTs and Pt–TiO_2_ NTs, were Au 4f and Pt 4f, respectively.

### 3.2. Antibacterial Agar Diffusion Test with Visible Light Irradiation

Figure 5 shows optical images and results of an antibacterial activity test. In Figure 5b, the initial CFU values of the control group of the three experimental conditions are not same, because different surface conditions provide the different bacterial adhesion abilities. However, the CFU values of Au–TiO_2_ NTs and Pt–TiO_2_ NTs under visible light irradiation were significantly reduced as compared to those without visible light irradiation, regardless of the various CFU values of the control groups (*p* < 0.05). In addition, the Au–TiO_2_ NTs and Pt–TiO_2_ NTs irradiated with 470 nm visible light showed the most effective antibacterial activity among all of the experimental conditions.

### 3.3. Biocompatibilty and Ostegenic Functionality Tests with Visible Light Irradiation

#### 3.3.1. Live/Dead Assay and MTT Assay

Figure 6 shows calcein AM and EthD-1 stained images and MTT assay results of hMSCs cultured on Au–TiO_2_ NTs and Pt–TiO_2_ NTs. There was no dramatic reduction of the number of viable hMSCs attached on the specimen from all experimental groups, as shown in Figure 6a. The results of the MTT assay show that there were no significant differences between the experimental groups after 24 and 48 h of incubation (*p* > 0.05). In addition, the values of all experimental groups were more than 70% of the control (blank) so that Au–TiO_2_ NTs and Pt–TiO_2_ NTs were proved to be biocompatible according to ISO 10993-5 [29]. As the first step for actual clinical application of the results of this study, the in vitro biocompatibility of the experimental specimens should be evaluated. The in vitro cytotoxicity tests of medical devices in dentistry are described in the ISO 7405 [30] and 10993-5 [29]. The MTT assay is one of in vitro cytotoxicity tests evaluating the biocompatibility of medical devices, and the cell viability values of experimental groups should be more than 70% of the control group to pass MTT assay according to ISO 10993-5. Therefore, Au–TiO_2_ NTs and Pt–TiO_2_ NTs can be biocompatible as a result of MTT assay.

#### 3.3.2. Micro Observation of hMSCs Morphology by FE-SEM

Figure 7 shows filopodia images of hMSCs cultures on Au–TiO_2_ NTs and Pt–TiO_2_ NTs before and after being irradiated for 30 min using 470 and 600 nm visible light. When 470 nm visible light was used on the hMSCs attached on Au–TiO_2_ NTs and Pt–TiO_2_ NTs, most filopodia of hMSCs were detached from the uppermost layer of the experimental specimen regardless of the coating materials. However, after 600 nm visible light irradiation for 30 min, the filopodia of hMSCs were more expanded and elongated than that of the control and 470 nm visible light irradiation groups. The phenomenon of cell detachment seems to be caused by the plasmonic photocatalysis of the Au–TiO_2_ NTs and Pt–TiO_2_ NTs expressed by 470 nm visible light. Therefore, the reactive oxygen species (ROS) and excited electrons that are generated by the plasmonic photocatalysis may interfere with the adhesion and growth of hMSCs [31]. On the other hand, 600 nm visible light is not related to the photocatalysis of titania. Thus, the filopodia attachment and extension of hMSCs are not supposed to be hindered by 600 nm visible light irradiation.

#### 3.3.3. Alkaline Phosphatase Activity Assay

Figure 8 shows the results of ALP activity after one and two weeks of cultivation of hMSCs with visible light irradiation. After one week of incubation, the ALP activity of Pt–TiO_2_ NTs was promoted by 600 nm visible light irradiation when compared to other experimental groups (*p* < 0.05). However, no significant improvement in ALP activity due to visible light irradiation was observed after a two-week incubation (*p* > 0.05). ALP is one of early stage markers when stem cells are differentiated into osteoblasts, so the ALP gene expression is increased during transformation of osteoprogenitor to osteoblast cells, which corresponds to the initial stage of the osteogenic differentiation of hMSCs [32,33]. When the differentiation of hMSCs reaches the osteocyte, the amount of ALP gene expression decreases significantly. Therefore, the ALP activity values of all experimental groups after 1 week of cultivation is higher than those after two weeks of cultivation due to the feature of the ALP expression, as shown Figure 8.

#### 3.3.4. Quantitative Real-Time PCR Assay

Figure 9 shows quantitative real-time PCR analysis data of ALP (after one week of incubation), OPN (after two weeks of incubation), and BSP (after two weeks of incubation) genes. ALP, OPN, and BSP are used as markers to monitor the extent of hMSCs osteogenic differentiation in vitro. As described in Section 3.3.3, the increase of ALP expression is observed during the initial stage of the osteogenic differentiation of hMSCs. OPN and BSP are mainly expressed on the stage of mature osteoblast and osteocytes, which corresponds to the late stage of hMSCs osteogenic differentiation [32,33]. Thus, we evaluated the ALP gene expression after one week of hMSCs culture, and OPN and BSP gene expression after two weeks of hMSCs culture. ALP and BSP gene expression values of Au-TiO_2_ NTs and PT-TiO_2_ NTs with 600 nm visible light irradiation were significantly higher than those with 470 nm visible light irradiation (*p* < 0.05). In addition, OPN and BSP gene expression values of Pt–TiO_2_ NTs after 600 nm visible light irradiation showed the highest values as compared to the other experimental groups under various visible light conditions.

## 4. Discussion

The fabrication of Au–TiO_2_ NTs and Pt–TiO_2_ NTs capable of antibacterial activity and osteogenic functionality by visible light irradiation was conducted by ion plasma sputtering. From the observation of FE-SEM and TEM images, most Au NPs and Pt NPs were deposited on the uppermost layer of the TiO_2_ NTs rather than in their inner surface. The morphology of Au and Pt NPs changed from a round to a rod shape with increasing deposition time of Au and Pt. When compared to the chemical coating of pre-synthesized Au NPs, the deposition of Au NPs and Pt NPs by ion plasma sputtering altered the shape of the deposited NPs on the TiO_2_ NTs due to the characteristic directional deposition of ion plasma sputtering.

In the four main peaks of diffuse UV–Vis–NIR spectra of Au–TiO_2_ NTs and Pt–TiO_2_ NTs, the absorbance peak around 350–400 nm relates to electron transition between the valence band and conduction band of TiO_2_ [34,35]. The absorbance peak around 450–500 nm is supposedly due to plasmonic photocatalysis from the LSPR effect of Au and Pt NPs deposited on TiO_2_ NTs. The Au and Pt NPs are thought to serve as an intermediate carrier of electron transition from the valence band to the conduction band of TiO_2_ NTs, thereby extending photocatalytic activity to the visible light region [36,37]. The two absorbance peaks detected at 550–650 nm and above 800 nm resulted from the photothermal scattering of the short and long axes of deposited Au and Pt NPs, respectively, and these results are strongly related to the morphology of Au and Pt NPs observed in the FE-SEM image [29,38,39,40]. From the results of FE-SEM, TEM, and diffuse UV–Vis–NIR spectra, Au sputtering for 1 min and Pt sputtering for 2 min were selected for further experiments to avoid agglomerated metal deposition and to maximize the visible light irradiation effects of Au–TiO_2_ NTs and Pt–TiO_2_ NTs at the 450–500 nm and 550–650 nm ranges, respectively.

From the XPS spectra of Au–TiO_2_ NTs and Pt–TiO_2_ NTs after 1 min of Au coating and 2 min of Pt coating, the main two peaks of Ti 2p shifted to lower binding energies as the Au and Pt NPs were coated on the surface of TiO_2_ NTs due to the reduction of Ti species. Additionally, the main peaks of O 1s were moved to lower binding energies due to the reduction of oxygen ions in the lattice with the Au and Pt NPs [41]. Thus, the lower shift of these major peaks is thought to cause incomplete Ti–O binding due to the embedment of Au and Pt [42].

As seen in the calcein AM and EthD-1-stained hMSCs images and MTT assay, all of the experimental groups were biocompatible according to ISO 10993-5. Additionally, from the results of the antibacterial test, 470 nm and 600 nm visible light irradiation were indicated to kill more bacteria on Au–TiO_2_ NTs and Pt–TiO_2_ NTs, and this phenomenon was strongly related to the plasmonic catalysis and photothermal scattering of Au and Pt NPs, as shown in the data of UV–Vis–NIR spectra (Figure 2).

From all of the results of ALP activity and the three osteogenic gene expressions (ALP, OPN, and BSP) of the real-time PCR assay, the osteogenic functionality of hMSCs cultured on Pt–TiO_2_ NTs under 600 nm visible light irradiation showed the most effective osteogenic enhancement of all the experimental groups. Thus, we investigated the correlation between various surface conditions, wavelength of visible light irradiation, and the adhesion of hMSCs to the surface of the specimen. From FE-SEM observation that is shown in Figure 7, when 600 nm visible light was used to irradiate hMSCs cultured on both Au–TiO_2_ NTs and Pt–TiO_2_ NTs, the filopodia of the hMSCs were observed to be well attached and elongated along the surface of the specimen. On the contrary, the filopodia of hMSCs under 470 nm visible light irradiation were found to be short or cut in the middle. Therefore, it was concluded that 600 nm visible light irradiation acted more positively on cell adhesion and proliferation of hMSCs than 470 nm visible light irradiation.

In terms of the visible light LED device used in this experiment, low-level laser therapy (LLLT) in the red and NIR region (600–1000 nm) is well known to play a major role in wound-healing, tissue repair, relief of inflammation pain, and so on [43,44,45,46]. LLLT, called photobiomodulation, is the application of light within the 600–1000 nm range to control the biological process of living cells. Thus, the 600 nm visible light LED device that was used in this study was expected to be synergic in terms of both osteogenic promotion and having an antibacterial effect.

Moreover, Pt–TiO_2_ NTs showed more osteogenic enhancement than Au–TiO_2_ NTs under 600 nm visible light irradiation and we speculate, from the data of diffuse UV–Vis–NIR spectra (Figure 3), that the wavelength of maximum intensity of Pt–TiO_2_ NTs was closer to 600 nm than that of Au–TiO_2_ NTs. Further investigation will be conducted to investigate this phenomenon.

When considering the practical applicability of dental clinics based on the results of this study, many dental prostheses made of titanium metal, such as implants, guided bone regeneration (GBR) [47], and orthodontic mini screws [48], are used in the dental clinic. Visible light cannot penetrate deep inside the tissue around the implant fixture, but it can easily reach the oral mucosa at the dental implant abutment area, so it is expected to be applicable for the quick healing of damaged soft tissue during the implant procedure. In addition, the oral mucosa is thin (<1 mm) enough to transmit 600 nm visible light (transmitting depth: 1.0 ≈ 2.0 mm) [49,50], the technique developed in this study is expected to be utilized for the rapid bone formation of the GBR titanium mesh implanted under the oral mucosa [51]. Moreover, the visible light-based osteogenic promotion technology is expected to significantly reduce the cost as compared to the conventional surgical techniques requiring highly expensive bone morphogenic proteins (BMPs) or growth factors to accelerate bone formation.

## 5. Conclusions

Within the limitation of this study, we confirmed that Au and Pt NPs deposited on TiO_2_ NTs by sputtering showed excellent antibacterial activity under 470 nm visible light irradiation due to the plasmonic photocatalysis from the LSPR effect of Au and Pt NPs that were attached to the surface of TiO_2_ NTs. Moreover, these Au and Pt NPs on TiO_2_ NTs promoted the osteogenic functionality of hMSCs under 600 nm visible light irradiation because of the combination effect of the photothermal scattering and the visible light LLLT. Therefore, we concluded that Au and Pt NPs deposited on TiO_2_ NTs could extend the limited use of TiO_2_ NTs from the UV to visible light region by LSPR and LLLT effects and, thereby, lubricate the development process of novel surface treatment techniques of implantable devices.

## Figures and Tables

**Figure 1 materials-13-03721-f001:**
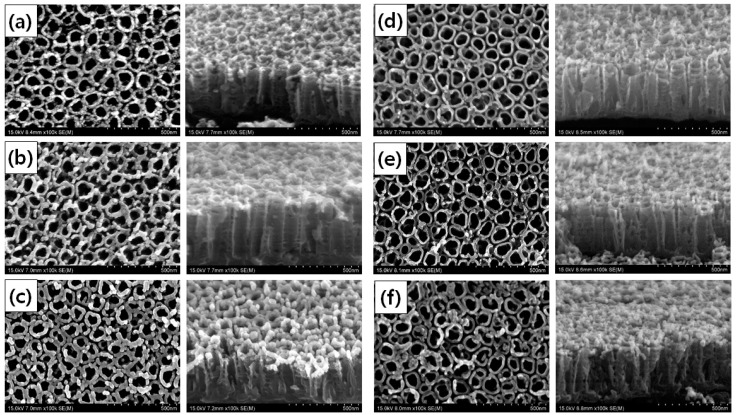
Field-emission scanning electron microscope (FE-SEM) images of 100 nm TiO_2_ nanotubes after (**a**) 1 min, (**b**) 2 min, and (**c**) 3 min of gold sputtering; and (**d**) 1 min, (**e**) 2 min, and (**f**) 3 min of platinum sputtering (top: plain view, bottom: oblique view.

**Figure 2 materials-13-03721-f002:**
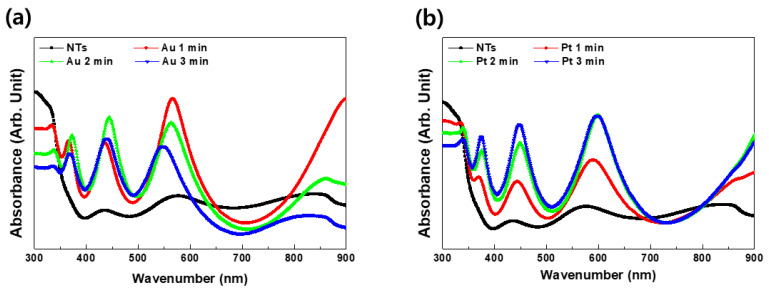
Diffuse UV–Vis–NIR spectra of (**a**) Au–TiO_2_ NTs and (**b**) Pt–TiO_2_ NTs after various sputtering periods.

**Figure 3 materials-13-03721-f003:**
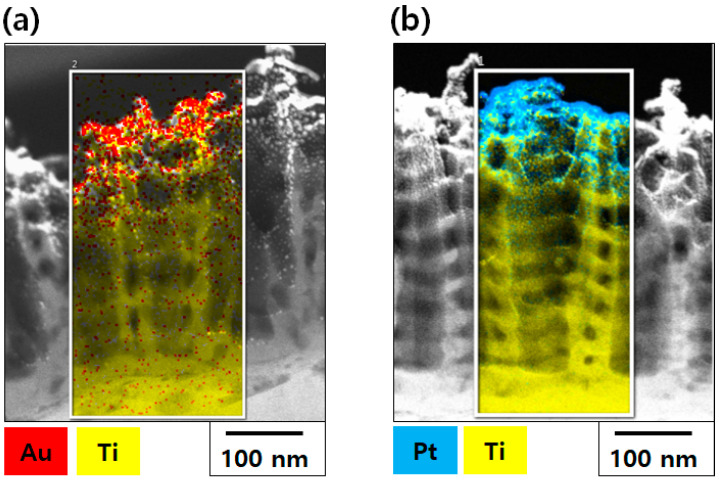
Transmission Electron Microscope (TEM) images of (**a**) Au–TiO_2_ NTs (1 min of Au sputtering) and (**b**) Pt–TiO_2_ NTs (1 min of Pt sputtering).

**Figure 4 materials-13-03721-f004:**
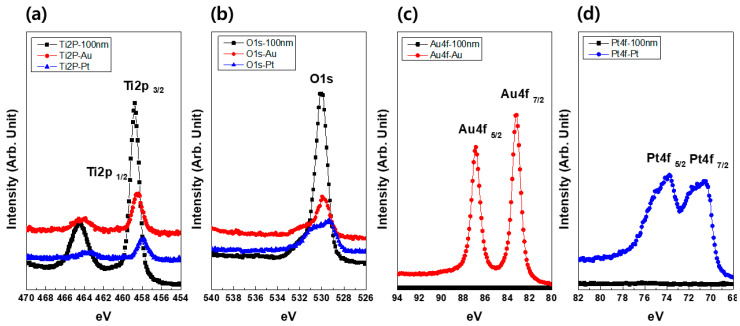
X-Ray Photoelectron Spectroscopy (XPS) spectra of (**a**) Ti2p, (**b**) O1s, (**c**) Au4f, and (**d**) Pt4f in Au–TiO_2_ NTs (1 min of Au coating) and Pt–TiO_2_ NTs (2 min of Pt coating).

**Figure 5 materials-13-03721-f005:**
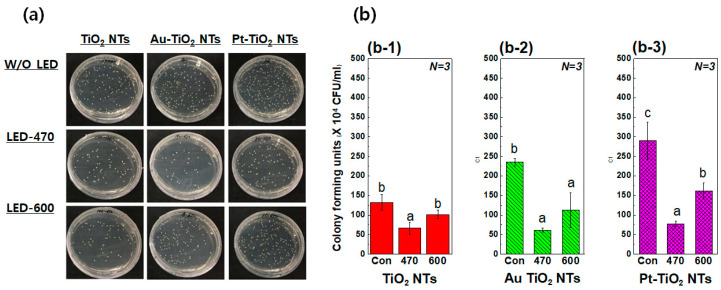
(**a**) Photographs of antibacterial Streptococcus aureus agar diffusion tests and (**b**) the results of colony forming units (CFUs) per unit volume of Streptococcus aureus cultured on (**b-1**) TiO_2_ NTs, (**b-2**) Au–TiO_2_ NTs, and (**b-3**) Pt–TiO_2_ NTs with or without 470 nm and 600 nm visible light irradiation (“Con”: No irradiation, “470”: 470 nm irradiation, “600”: 600 nm irradiation) (in each graph, entries with the same lowercase letters were not significantly different as determined by one-way ANOVA at α = 0.05).

**Figure 6 materials-13-03721-f006:**
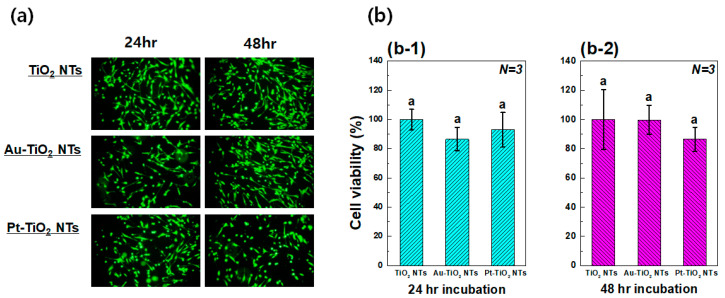
(**a**) Calcein AM and EthD-1 stained images and (**b**) MTT assay results of Human mesenchymal stem cells (hMSCs) cultured on TiO_2_ NTs, Au–TiO_2_ NTs and Pt–TiO_2_ NTs after (**b-1**) 24 h and (**b-2**) 48 h incubation (in each graph, entries with the same lowercase letters were not significantly different as determined by one-way ANOVA at α = 0.05).

**Figure 7 materials-13-03721-f007:**
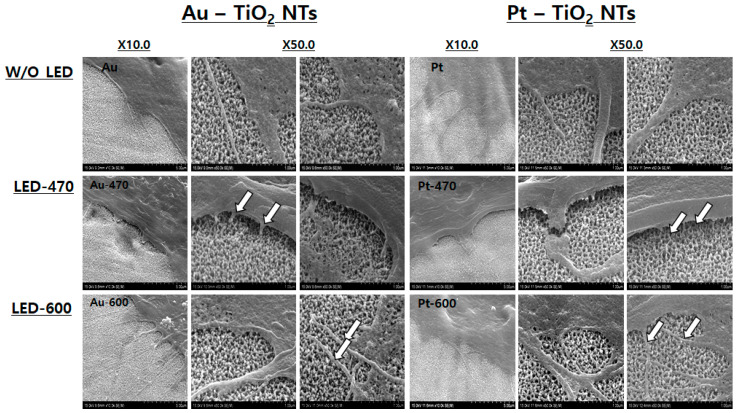
Filopodia images of hMSCs cultured on Au–TiO_2_ NTs and Pt–TiO_2_ NTs before and after 470 nm and 600 nm visible light irradiation (as observed by FE-SEM).

**Figure 8 materials-13-03721-f008:**
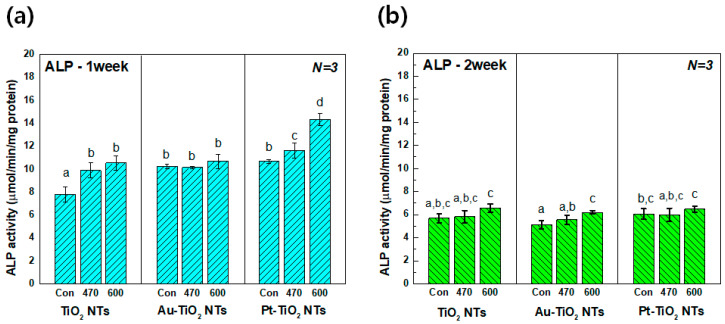
Alkaline Phosphatase (ALP) activity of hMSCs cultured on TiO_2_ NTs, Au–TiO_2_ NTs, and Pt–TiO_2_ NTs after (**a**) 1 week and (**b**) two weeks of incubation (“Con”: No irradiation, “470”: 470 nm irradiation, “600”: 600 nm irradiation) (in each graph, entries with the same uppercase and lowercase letters were not significantly different as determined by one-way ANOVA at α = 0.05).

**Figure 9 materials-13-03721-f009:**
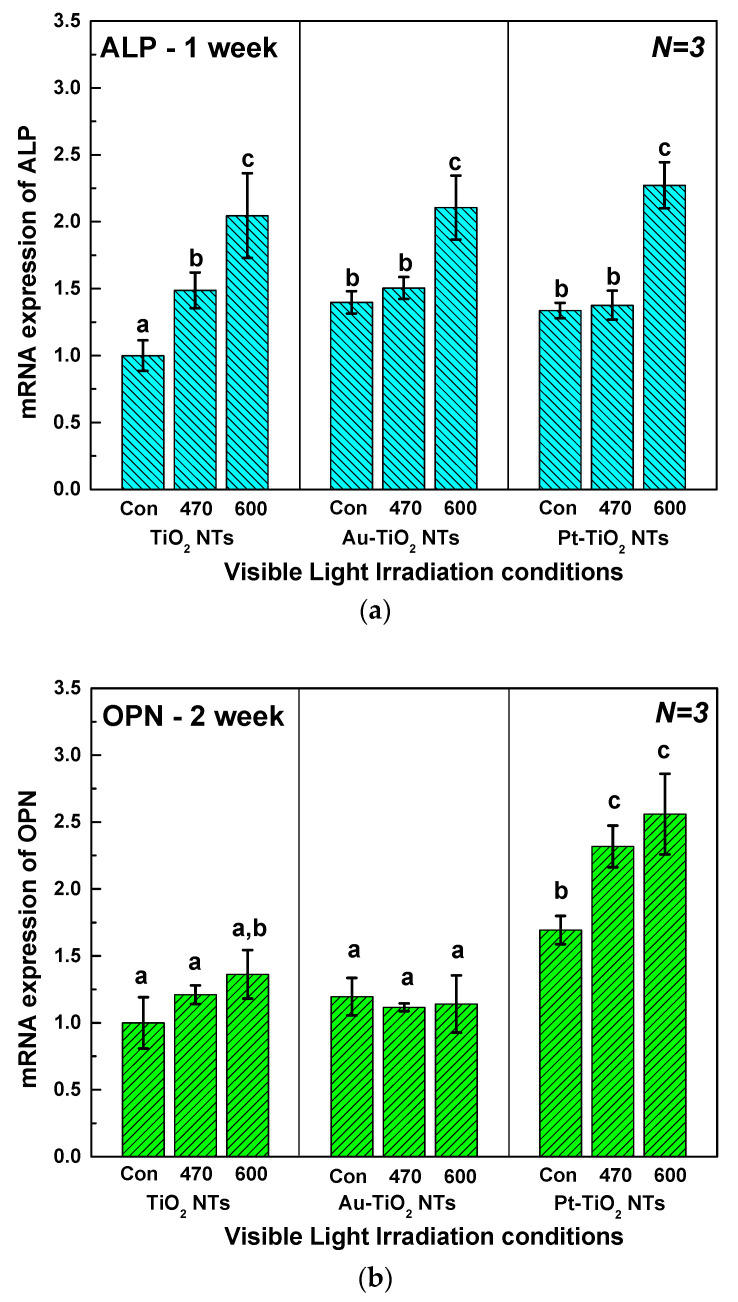
Quantitative real-time PCR analysis for (**a**) ALP, (**b**) OPN, and (**c**) BSP of TiO_2_ NTs, Au–TiO_2_ NTs and Pt–TiO_2_ NTs under 470 and 600 nm visible light irradiation (“Con”: No irradiation, “470”: 470 nm irradiation, “600”: 600 nm irradiation) (in each graph, entries with the same lowercase letters were not significantly different as determined by one-way ANOVA at α = 0.05).

**Table 1 materials-13-03721-t001:** The diameter, height, and aspect ratio of Au and Pt nanoparticles deposited on TiO_2_ nanotubes with various sputtering time.

Metal Target	Sputtering Time (min)	Diameter (nm)	Height (nm)	Aspect Ratio
Au	1	9.36 ± 1.88	8.77 ± 1.90	1:0.94
	2	13.05 ± 2.22	11.79 ± 2.41	1:0.90
	3	18.29 ± 6.44	30.77 ± 3.28	1:1.68
Pt	1	15.35 ± 4.31	18.65 ± 3.75	1:1.22
	2	20.72 ± 5.21	30.42 ± 6.01	1:1.47
	3	24.23 ± 4.40	48.24 ± 6.41	1:1.99

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
