# Peer review of "Visible Light-Enhanced Antibacterial and Osteogenic Functionality of Au and Pt Nanoparticles Deposited on TiO2 Nanotubes"

_materials, 2020, doi:10.3390/ma13173721_

Round 1
Reviewer 1 Report
The paper focuses on Au-deposited TiO2 NTs and Pt-deposited TiO2 NTs prepared by ion plasma sputtering and the evaluation of their antibacterial and osteogenic performance. A range of experimental techniques were used including: Field Emission Scanning Electron Microscope, Transmission Electron Microscope, X-ray photoelectron spectroscopy, UV-Vis-NIR spectroscopy, antibacterial agar diffusion tests, live/dead assay, MTT assay, alkaline phosphatase activity assay and quantitative real-time PCR assay The systems show excellent antimicrobial activity under visible radiation and enhanced osteogenic functionality of human mesenchymal stem cells under 600 nm irradiation, thus carrying promise for the development of advanced implantable devices. The work could be of interest for the readers of the journal.
The authors make reference to a number of previous close-related studies, including few of their own reports. In that sense, the novelty of their work should be articulated more clearly both from a scientific and a technological viewpoint.
The authors should comment on the limitations and the cost of the systems compared to available technologies.
A performance evaluation under different conditions and as a function of time would be welcome.
Author Response
Reviewer 1
Comments to the Author
The paper focuses on Au-deposited TiO2 NTs and Pt-deposited TiO2 NTs prepared by ion plasma sputtering and the evaluation of their antibacterial and osteogenic performance. A range of experimental techniques were used including: Field Emission Scanning Electron Microscope, Transmission Electron Microscope, X-ray photoelectron spectroscopy, UV-Vis-NIR spectroscopy, antibacterial agar diffusion tests, live/dead assay, MTT assay, alkaline phosphatase activity assay and quantitative real-time PCR assay The systems show excellent antimicrobial activity under visible radiation and enhanced osteogenic functionality of human mesenchymal stem cells under 600 nm irradiation, thus carrying promise for the development of advanced implantable devices. The work could be of interest for the readers of the journal.
The authors make reference to a number of previous close-related studies, including few of their own reports. In that sense, the novelty of their work should be articulated more clearly both from a scientific and a technological viewpoint.
è Thank you for your comment. In our previous study, antibacterial agents were eluted by the visible light or near IR light irradiation, resulting in an antibacterial effect. However, in this study, the antibacterial effect was expressed by the optical properties of the titania nanotubes coated with noble metals and visible light irradiation without antibacterial agents. Also, no one has conducted any studies on the evaluation of osteogenic functionality using the combination of noble metal coated titania nanotubes and visible light. So, it can be said that there is a clear difference between the previous study and current study, and the novelty of this current study is high enough.
è I have revised the sentence summarizing our previous studies and explaining the novelty of our current study. (Line 57)
|
In previous studies, we found four major absorbance peaks in the ultraviolet (UV), visible, and near infrared (NIR) regions and evaluated the visible–NIR laser-mediated release of antibacterial agent and the accompanying antibacterial activity of Au-coated TiO2 nanotubes (NTs) [23, 24]. There are few studies relating to antimicrobial activity using plasmonic photocatalysis and the photothermal scattering effect of noble metal-coated TiO2 NTs without antibacterial agents [25-27]. Also, the research relating to the combined effect of visible light and noble metal coated TiO2 NTs on the osteogenic functionality of human mesenchymal stem cells was not conducted in detail. |
The authors should comment on the limitations and the cost of the systems compared to available technologies.
è Thank you for the good comment. I put the sentence explaining (1) the practical applicability of dental practices based on the results of this study and (2) the cost of the technology derived from this study compared to the existing surgical techniques. (Line 371)
|
Considering the practical applicability of dental clinics based on the results of this study, many dental prostheses made of titanium metal such as implants, guided bone regeneration (GBR) [47], and orthodontic mini screws [48] are used in the dental clinic. Visible light cannot penetrate deep inside the tissue around the implant fixture, but it can easily reach the oral mucosa at the dental implant abutment area, so it is expected to be applicable for the quick healing of damaged soft tissue during the implant procedure. In addition, the oral mucosa is thin (<1 mm) enough to transmit 600 nm visible light (transmitting depth: 1.0 ~ 2.0 mm) [49, 50], the technique developed in this study is expected to be utilized for the rapid bone formation of the GBR titanium mesh implanted under the oral mucosa [51]. Moreover, the visible light-based osteogenic promotion technology is expected to significantly reduce the cost compared to the conventional surgical techniques requiring highly expensive bone morphogenic proteins (BMPs) or growth factors to accelerate bone formation. |
è I have added the phrase the limitation of this study in the section of 5. Conclusion.
|
Within the limitation of this study, we confirmed…. |
A performance evaluation under different conditions and as a function of time would be welcome.
è Thank you for the good comment. We will conduct the research on the antibacterial activity and osteogenic functionality of noble metal coated titania nanotubes with various wave numbers and irradiation periods of visible light.

Reviewer 2 Report
The manuscript is well written, and the scientific theory is well thought out. The authors used a material platform to promote gene expression while creating an antibiotic surface. This is a combined optogenetic approach to cell differentiation that is controlled by physical plasmonic surface treatment.
However, I do have a few comments and questions:
Figure 5. Why is the control colony forming units different for each condition? Shouldn’t the starting CFU be the same. When you see the number of CFUs for TiO2 NTs, Au TiO2 NTs, and Pt TiO2 NTs irradiated at 470 nm and 600 nm, the CFU counts are the same. Shouldn’t they be significantly different. You’re comparing it to the control but the control should be the same for each one. Where was the TiO2 NT sheet?
Figure 6. What is significant of the values of all the experimental groups were 70% of the control? You have a control stated at 100% and the hMSCs showed a lower percentage on each surface. The results are significant but section 3.3.1 needs more explanation.
Figure 7. Can you explain why at 470 nm the filopodia are not attached to the surface but at 600 nm the filopodia are expanded and elongated?
Figure 8. Is there any other explanation to see a decrease in ALP activity? You’re seeing a decrease of greater than 30%, could this be due to age and not by the irradiation? Section 3.3.3 needs more explanation.
Section 3.3.4 needs more explanation of why the genes were selected. Is there any way to explain ALP activity with filopodia explanation.
One major question is since this surface is to regenerate bone how would you get the visible light to the material surface? This is a major drawback of the work. It is very interesting and well thought out, but in practice it would be hard to conduct and complete.
Author Response
Reviewer 2 comments
The manuscript is well written, and the scientific theory is well thought out. The authors used a material platform to promote gene expression while creating an antibiotic surface. This is a combined optogenetic approach to cell differentiation that is controlled by physical plasmonic surface treatment.
However, I do have a few comments and questions:
Figure 5. Why is the control colony forming units different for each condition? Shouldn’t the starting CFU be the same. When you see the number of CFUs for TiO2 NTs, Au TiO2 NTs, and Pt TiO2 NTs irradiated at 470 nm and 600 nm, the CFU counts are the same. Shouldn’t they be significantly different. You’re comparing it to the control, but the control should be the same for each one. Where was the TiO2 NT sheet?
è Thank you for the good comment. It is reasonable that different surface conditions give the different degrees of the bacterial attachment to the surface resulting in different starting CFU value. Generally, the CFU value is expressed as a percentage by dividing the CFU value of the experimental group into the CFU value of the control group. However, since the control groups used in the three conditions are not the same, the bacterial adherence rates of the three control groups are also different each other. So, the measured CFU values of control groups (without visible light irradiation) and experimental groups (470 and 600 nm visible light irradiation) were recorded on the graph in order to explain (1) the difference in the bacterial surface affinity according to the surface conditions and (2) the antibacterial activity mediated by visible light irradiation.
è I put the sentence explaining why the starting CFU values of the controls. (Line 226)
|
In Figure 5(b), the initial CFU values of the control group of the three experimental conditions are not same because different surface conditions provide the different bacterial adhesion abilities. However, the CFU values of Au–TiO2 NTs and Pt–TiO2 NTs under visible light irradiation were significantly reduced compared to those without visible light irradiation regardless of the various CFU values of the control groups (P<0.05). |
è On the condition of an antibacterial effect experiment using a specimen loaded with an antibacterial drug, it is common to place the specimen on a 100 mm culture dish, inoculate the bacteria, and measure the area of the bacteria killing zone after specific culture periods. However, this study attempted to observe the antibacterial activity of the test specimen by light irradiation without antibacterial drugs. In short, after visible light irradiation, 100 μL of the bacteria cultured on the specimen was collected and diluted in 1x phosphate buffered saline (PBS) solution. Then, 100 μL of the diluted bacteria solution was inoculated on a 100 mm diameter agar plate and incubated for 24 h at 37° C. After 24 h of incubation, the resulting CFU per unit volume was measured by visually inspecting the agar plate. This is the reason why there are no titania nanotube specimens on the 100 mm petri dishes in Figure 5(a). The detailed procedure of the antibacterial activity test was described in 2.3 Antibacterial activity test.
è In Figure 5(b), the number of CFUs for TiO2 NTs, Au TiO2 NTs, and Pt TiO2 NTs irradiated at 470 nm and 600 nm are not the same. In terms of statistical analysis, the same lower-case letters were not significantly different as determined by one-way ANOVA in each graph. Figure 5(b) is not one graph, but three graphs, and statistical analysis were performed on each graph.
è To clarify the explanation of Figure 5(a) and (b), the abbreviation letters were revised, and more explanations were added to the caption of Figure 5.
Figure 6. What is significant of the values of all the experimental groups were 70% of the control? You have a control stated at 100% and the hMSCs showed a lower percentage on each surface. The results are significant but section 3.3.1 needs more explanation.
è Thank you for the good comment. As the first step for the actual clinical application of the results of this study, the biocompatibility of the experimental specimens should be evaluated. The evaluations of the in vitro biological safety of medical devices in dentistry are described in the ISO 7405 and ISO 10993-5. Among them, MTT assay is described in ISO 10993-5, and the 70% of cell viability values resulted from MTT assay is the minimum required values according to ISO 10993-5. Although the results of the experimental group showed a somewhat lower value than that of the control group, no statistically significant difference was found. In addition, all experimental groups were biocompatible because the cell viability values of the experimental groups exceeded the minimum required value of the international standard.
è I put the sentence explaining the meaning of 70% cell viability at 3.3.1. (line 250)
|
As the first step for actual clinical application of the results of this study, the in vitro biocompatibility of the experimental specimens should be evaluated. The in vitro cytotoxicity tests of medical devices in dentistry are described in the ISO 7405 [30] and 10993-5. MTT assay is one of in vitro cytotoxicity tests evaluating the biocompatibility of medical devices, and the cell viability values of experimental groups should be more than 70% of the control group to pass MTT assay according to ISO 10993-5. Therefore, Au–TiO2 NTs and Pt–TiO2 NTs can be biocompatible as a result of MTT assay. |
Figure 7. Can you explain why at 470 nm the filopodia are not attached to the surface but at 600 nm the filopodia are expanded and elongated?
è Thank you for the good comment. Research is currently underway to clarify this phenomenon. The mechanism currently inferred is that plasmonic photocatalysis expressed in visible light at 470 nm is a photocatalytic phenomenon expressed on the surface of titania nanotubes by visible light. Therefore, the reactive oxygen species (ROS) generated on the surface may interfere with the adhesion and growth of bacteria. On the other hand, the visible light of 600 nm is irrelevant to the photocatalytic phenomenon, so the cell adhesion is seemed to be performed the same as usual.
è I put the sentence explaining the cell detachment on the condition of 470 nm visible light irradiation. (line 270)
|
The phenomenon of cell detachment seems to be caused by the plasmonic photocatalysis of the Au–TiO2 NTs and Pt–TiO2 NTs expressed by 470 nm visible light. Therefore, the reactive oxide species (ROS) and excited electrons generated by the plasmonic photocatalysis may interfere with the adhesion and growth of hMSCs [31]. On the other hand, 600 nm visible light is not related to the photocatalysis of titania. Thus, the filopodia attachment and extension of hMSCs are not supposed to be hindered by 600 nm visible light irradiation. |
Figure 8. Is there any other explanation to see a decrease in ALP activity? You’re seeing a decrease of greater than 30%, could this be due to age and not by the irradiation? Section 3.3.3 needs more explanation.
è Thank you for the good comment. ALP is one of early stagmarkers when stem cells are differentiated into osteoblasts. Also, the increase of ALP expression is observed during transformation of osteoprogenitor to pre-osteoblasts cells. Therefore, in Figure 8, it is natural that the discovery of ALP in one week of culture is large and decreases after that.
è I put the sentence explaining the trend of ALP activity after 1 and 2 weeks of hMSCs incubation. (Line 285)
|
ALP is one of early markers when stem cells are differentiated into osteoblasts, so ALP gene expression is increased during transformation of osteoprogenitor to osteoblast cells, which corresponds to the initial stage of the osteogenic differentiation of hMSCs [32, 33]. When the differentiation of hMSCs reaches the osteocyte, the amount of ALP gene expression decreases significantly. Therefore, the ALP activity values of all experimental groups after 1 week of cultivation is higher than those after 2 weeks of cultivation due to the feature of the ALP expression as shown Figure 8. |
Section 3.3.4 needs more explanation of why the genes were selected. Is there any way to explain ALP activity with filopodia explanation.
è Thank you for the good comment. Alkaline phosphatase (ALP), osteopontin (OPN), and bone sialoprotein (BSP) are used as markers to monitor the extent of MSCs osteogenic differentiation in vitro. ALP is an early marker of osteoblast, and the increase of ALP expression is observed during transformation of osteoprogenitor to pre-osteoblasts cells. OPN and BSP are also markers expressed on the stage of mature osteoblast and osteocytes. Thus, we evaluated the ALP gene expression after 1 week of hMSCs culture, and OPN and BSP gene expression after 2 weeks of hMSCs culture.
è I put the sentence explaining the selection of ALP, OPN, and BSP genes in detail. (Line 300)
|
ALP, OPN, and BSP are used as markers to monitor the extent of hMSCs osteogenic differentiation in vitro. As described in section of 3.3.3., the increase of ALP expression is observed during the initial stage of the osteogenic differentiation of hMSCs. OPN and BSP are mainly expressed on the stage of mature osteoblast and osteocytes, which corresponds to the late stage of hMSCs osteogenic differentiation [32, 33]. Thus, we evaluated the ALP gene expression after 1 week of hMSCs culture, and OPN and BSP gene expression after 2 weeks of hMSCs culture. |
è In terms of the relation between ALP activity and filopodia explanation, we could not find the correlation between the two, and further investigation will be performed.
One major question is since this surface is to regenerate bone how would you get the visible light to the material surface? This is a major drawback of the work. It is very interesting and well thought out, but in practice it would be hard to conduct and complete.
è Thank you for the good comment. Many dental prostheses made of titanium metal such as implants, guided bone regeneration, and orthodontic mini screws are used in the dental clinic. In the case of the dental implant, visible light cannot be transferred into the deep inside of the implant fixture but can be reached to the abutment site of dental implant for the fast healing of the damaged soft tissue around implantation site. In addition, since the oral mucosa is thin (<1 mm) and 600 nm visible light can be sufficiently transmitted up to 2.0 mm, it is considered that it can be sufficiently utilized for the rapid bone formation of the titanium mesh for GBR.
è I put the sentence explaining the practical applicability of dental clinics based on the results of this study and stating the cost of the system developed by our study compared to available technologies. (Line 371)
|
Considering the practical applicability of dental clinics based on the results of this study, many dental prostheses made of titanium metal such as implants, guided bone regeneration (GBR) [47], and orthodontic mini screws [48] are used in the dental clinic. Visible light cannot penetrate deep inside the tissue around the implant fixture, but it can easily reach the oral mucosa at the dental implant abutment area, so it is expected to be applicable for the quick healing of damaged soft tissue during the implant procedure. In addition, the oral mucosa is thin (<1 mm) enough to transmit 600 nm visible light (transmitting depth: 1.0 ~ 2.0 mm) [49, 50], the technique developed in this study is expected to be utilized for the rapid bone formation of the GBR titanium mesh implanted under the oral mucosa [51]. Moreover, the visible light-based osteogenic promotion technology is expected to significantly reduce the cost compared to the conventional surgical techniques requiring highly expensive bone morphogenic proteins (BMPs) or growth factors to accelerate bone formation. |

Round 2
Reviewer 1 Report
The revised manuscript is improved and can be published as it stands now.